# Novel Approach for Identification of Basic and Effective Reproduction Numbers Illustrated with COVID-19

**DOI:** 10.3390/v15061352

**Published:** 2023-06-11

**Authors:** Tchavdar T. Marinov, Rossitza S. Marinova, Radoslav T. Marinov, Nicci Shelby

**Affiliations:** 1Department of Natural Sciences, Southern University at New Orleans, 6801 Press Drive, New Orleans, LA 70126, USA; nicci.shelby@sus.edu; 2Department of Mathematical & Physical Sciences, Concordia University of Edmonton, 7128 Ada Boulevard, Edmonton, AB T5B 4E4, Canada; rossitza.marinova@concordia.ab.ca; 3Department Computer Science, Varna Free University, 9007 Varna, Bulgaria; 4Rescale, 33 New Montgomery Street, Suite 950, San Francisco, CA 94105, USA; rmarinov@rescale.com

**Keywords:** SIR model, time-dependent parameters, reproduction numbers, inverse problem, infection and recovery rates, data mining, infectious disease modeling, epidemic dynamics

## Abstract

This paper presents a novel numerical technique for the identification of effective and basic reproduction numbers, $R_{e}$ and $R_{0}$, for long-term epidemics, using an inverse problem approach. The method is based on the direct integration of the SIR (Susceptible–Infectious–Removed) system of ordinary differential equations and the least-squares method. Simulations were conducted using official COVID-19 data for the United States and Canada, and for the states of Georgia, Texas, and Louisiana, for a period of two years and ten months. The results demonstrate the applicability of the method in simulating the dynamics of the epidemic and reveal an interesting relationship between the number of currently infectious individuals and the effective reproduction number, which is a useful tool for predicting the epidemic dynamics. For all conducted experiments, the results show that the local maximum (and minimum) values of the time-dependent effective reproduction number occur approximately three weeks before the local maximum (and minimum) values of the number of currently infectious individuals. This work provides a novel and efficient approach for the identification of time-dependent epidemics parameters.

## 1. Introduction

Infectious diseases have become a particularly important subject of study in recent years due to the COVID-19 epidemic. Daily reports from across the world have enabled scientists to collect data for research into new approaches for studying infectious diseases. Early identification of outbreaks is of major concern to public health authorities [1,2,3,4].

Key parameters of epidemic spread are the basic reproduction number, $R_{0}$ and the effective reproduction number, $R_{e}$. These measure the severity of the infection.

The *basic reproduction number* $R_{0}$ is defined in epidemiology as the average number of secondary infections caused by a single infected person in a susceptible population, as found in [5,6,7]. It is used to measure the transmission potential of a disease;The *effective reproduction number* $R_{e}$ is defined as the product of the basic reproduction number and the fraction of the host population that is susceptible. It estimates the rate of spread of an epidemic.

In [8] and other works [6,7,9,10,11,12], the focus was on the basic reproduction number, whereas the authors of [13] estimated the effective reproduction number. They applied their method to the spread of COVID-19 in municipalities within Rio de Janeiro, Brazil. Alvarez et al. [14] proposed a variational technique for inverting the renewal equation, based on two formulations of the renewal equation presented in [15,16], to estimate the daily reproduction number.

The SIR model, one of the simplest models for the spread of an infectious disease, was proposed in [17] in 1927. It categorizes individuals as **S**usceptible, **I**nfectious, and **R**ecovered (SIR). The SIR model assumes that infectious individuals transition to recovered and become immune, and the recovered group includes deceased individuals. SIR models displayed compelling results, especially during the early stages of the pandemic [18,19,20,21,22,23,24,25,26,27,28,29,30,31].

Recent works have implemented various SIR-type models to investigate the COVID-19 infectious disease dynamics [10,32,33,34]. Variations of the basic SIR model applied to COVID-19 data include SIS (S for susceptible, no immunity upon recovery), SIRD (D for deceased—recovered and dead considered separately), SIRV (V for vaccinated), SEIR (E for exposed), MSEIR (M for maternal/passively immune) [35], or MSEIR (meso-scale SEIR) [36]. Researchers have used SIR and SEIR-based models with vaccination to overcome the limitations of the conventional SIR model when applied to the COVID-19 epidemic [37,38,39]. Zhao et al. [38] applied statistical methods to estimate parameters.

The present work proposes a novel approach for the identification of the time-dependent reproduction numbers. It extends the SIR-type model by assuming time-dependent infection and recovery rates [40,41], which are required for the estimation of the reproduction numbers. The method is based on direct integration of the SIR system and an inverse problem approach for solving the resulting equation for the ratio of the infection and recovery rates. The evaluation of the time-dependent reproduction numbers over long time periods gives a tool for investigating the wave formation of the infection curve for the COVID-19 epidemic. This study examines and shows how the time-dependent effective reproduction number compares to the number of infectious individuals over time and gives a novel approach for predicting epidemic waves.

## 2. The SIR Model for the Spread of an Infectious Disease

The notations in the SIR model include S(t)—the number of *susceptible*, I(t)—the number of *infectious*, R(t)—the number of *removed* individuals, and *t*—time. The total population N=S(t)+I(t)+R(t) is considered constant in the above equations.

The *infection rate* β (0<β<1) is the probability that a random infectious person infects a random susceptible person. A major approximation here is the assumption that the population under study is well-mixed, so that every person has an equal probability of coming into contact with every other person. The *recovery rate*γ (0<γ<1) gives the probability that an infectious person recovers.

The SIR model consists of the following system of differential equations: (1)$$\begin{array}{cc} \frac{dS(t)}{dt} & =-\beta S(t)I(t); \end{array}$$(2)$$\begin{array}{cc} \frac{dI(t)}{dt} & =\beta S(t)I(t)-\gamma I(t); \end{array}$$(3)$$\begin{array}{cc} \frac{dR(t)}{dt} & =\gamma I(t). \end{array}$$

Figure 1 shows the diagram for the SIR model, corresponding to the system (Equation 1)–(3). 

The SIR model assumes that the removed individuals are no longer susceptible or infectious. The number of cases of people recovered from COVID-19 who are re-infected at the present moment is very limited, and the rate cannot be estimated; thus, the possibility of reinfection is not taken into account. Specific details on the assumptions and the adaptive SIR (A-SIR) model can be found in [40,41].

### 2.1. Integrating the SIR System

Equations (Equation 1) and (Equation 2) do not depend on the function R(t). Therefore, the SIR system can be viewed as consisting of two parts: (i) Equations (Equation 1) and (Equation 2), and (ii) Equation (Equation 3). If the function I(t) is known, the last Equation (Equation 3) can be solved by direct integration.

Let the coefficients β and γ be constant, and S(t)>0, I(t)>0, β≠0, γ≠0.

From Equations (Equation 1) and (Equation 2) follows the ordinary differential equation below, connecting the functions *I* and *S* (see also [42]):
(4)$$\frac{dI}{dS}=\frac{\beta SI-\gamma I}{-\beta SI},$$
namely,
(5)$$\frac{dI}{dS}=-1+\frac{\gamma }{\beta S}\cdot$$Integrating with respect to *S* yields the following general solution to Equation (Equation 5):
(6)$$I=-S+\frac{\gamma }{\beta }\ln S+C,$$
representing *I* as a function of *S*;Another way to integrate the system is by excluding dt from Equations (Equation 1) and (Equation 3). This gives the following ordinary differential equation connecting the functions S(t) and R(t):
(7)$$\frac{dS}{dR}=-\frac{\beta }{\gamma }S.$$The general solution to Equation (Equation 7) is then as follows:
(8)$$\frac{\beta }{\gamma }R\ln S+C=0,$$
representing *R* as a function of *S*.

### 2.2. The Effective and Basic Reproduction Numbers $R_{e}$ and $R_{0}$

The conditions under which an epidemic occurs present an important question. An epidemic occurs if the number of infected individuals I(t) is increasing [12,42]. This means that the rate of change of the infectious, $\frac{dI}{dt}$, is positive. From Equation (Equation 2), the following is implied:$$\frac{dI(t)}{dt}=\beta S\left(t\right)I\left(t\right)-\gamma I\left(t\right)>0,$$
which results in the following condition:(9)$$R_{e}\left(t\right)=\frac{\beta S(t)}{\gamma }>1.$$

The parameter $R_{e}\left(t\right)$ is called the *effective reproduction number* (also referred to as *ratio* or *rate*) for a given disease. From a statistical point of view, the effective reproduction number is the expected number of secondary cases produced, in a completely susceptible population, by a typical infectious individual.

Another important characteristic of an infectious disease, *the basic reproduction number (ratio, rate)* $R_{0}$, can be computed as the ratio of the known rates β and γ over time:(10)$$R_{0}=\frac{\beta N}{\gamma },$$
where *N* is the size of the total population.

The basic reproduction number, $R_{0}$, is used for answering two important questions, the first being what fraction of the population would become ill should an epidemic occur, and the second being what is the minimum fraction of the population that must be vaccinated in order to prevent an epidemic.

Diseases with smaller values of $R_{0}=\frac{\beta N}{\gamma }$ are easier to eradicate than those with larger values $R_{0}$, since a population can acquire herd immunity with a smaller fraction of the population vaccinated. For example, smallpox with $R_{0}\approx 4$ has been eradicated worldwide, while measles with $R_{0}\approx 14$ still has occasional outbreaks.

### 2.3. The Direct and the Inverse Problem

The initial value problem consisting of the system (Equation 1)–(Equation 3) with known constant coefficients β and γ, along with proper initial conditions derived from the given data, constitutes the direct problem. In reality, for a new disease, the values of the parameters β and γ are unknown. Finding these parameters involves solving an inverse problem for determining the coefficients and functions from the available data. The inverse problem for the estimation of the constants β and γ in the classical SIR model is solved in [43].

### 2.4. Time-Dependent Infection and Recovery Rates and Reproduction Numbers

The original SIR model assumes that the infection and recovery rates are constants. Equations (Equation 1)–(Equation 3), with proper initial conditions, allow the determination of I(t), S(t), and R(t), if the coefficients β and γ are known constants.

However, in the case of a pandemic, the rates may vary over time; hence, β=β(t) and γ=γ(t). Multiple factors can cause the rates to change over time. In the case of COVID-19, examples include social distancing, government restrictions, and preventive treatments.

Therefore, the basic reproduction number can be assumed to be a function of time and can be defined:(11)$$R_{0}\left(t\right)=\frac{\beta (t)}{\gamma (t)}\;N,$$
with the effective reproduction number as follows:(12)$$R_{e}\left(t\right)=\frac{\beta (t)}{\gamma (t)}\;S\left(t\right).$$

An algorithm for solving the inverse problem (Equation 1) and (Equation 2), under the condition that β=β(t) and γ=γ(t) are functions of time, is given in [40]. Knowing β=β(t) and γ=γ(t) allows us to estimate the reproduction numbers. The present work focuses on alternative methods for directly identifying the reproduction numbers from a given dataset, as well as comparisons of the different approaches.

## 3. The Inverse Problem for the Time-Dependent Reproduction Numbers

Let the values of S(t) and I(t) at some time moments, $\nu_{1}$, $\nu_{2}$, …, $\nu_{m}$, shown in Figure 2, be known and given by the following:(13)$$S\left(\nu_{l}\right)=\sigma_{l},\;I\left(\nu_{l}\right)=\lambda_{l},\;l=1,2,\ldots ,m.$$

If the population *N* is constant, the values of R(t) at time moments $\nu_{l}$ can be obtained:(14)$$R\left(\nu_{l}\right)=\rho_{l}=N-\sigma_{l}-\lambda_{l}.$$

Assuming *N* and S(t) are available, the time-dependent reproduction numbers can be determined after the estimation of the ratio γ(t)/β(t).

The following two approaches for estimating the constant γ/β in the differential Equation (Equation 5), or β/γ in (Equation 7), at time moments $\nu_{1},\nu_{2},\ldots ,\nu_{m}$ are used in this work:(i)By solving the boundary value problem (BVP) for (Equation 5) with the following boundary conditions:
(15)$$\begin{array}{c} S\left(\nu_{l-1}\right)=\sigma_{l-1},\;I\left(\nu_{l-1}\right)=\lambda_{l-1}, \end{array}$$
(16)$$\begin{array}{c} S\left(\nu_{l}\right)=\sigma_{l},\;I\left(\nu_{l}\right)=\lambda_{l}, \end{array}$$
for l=2,3,…,m;(i’)By solving the BVP for (Equation 7) with the following boundary conditions:
(17)$$\begin{array}{c} S\left(\nu_{l-1}\right)=\sigma_{l-1},\;R\left(\nu_{l-1}\right)=\rho_{l-1}, \end{array}$$
(18)$$\begin{array}{c} S\left(\nu_{l}\right)=\sigma_{l},\;R\left(\nu_{l}\right)=\rho_{l}, \end{array}$$
for l=2,3,…,m;(ii)By using the least-squares method (LSM) for estimating the parameter γ/β in (Equation 6);(ii’)By using the LSM for estimating the unknown parameter β/γ in (Equation 8).

### 3.1. Estimating γ/β as a Solution of BVP for Equation (Equation 5)

The solution (Equation 6) to Equation (Equation 5) can be written as shown:(19)$$\frac{\gamma }{\beta }\ln S+C=S+I.$$

Equation (Equation 19) and the conditions (Equation 15) and (Equation 16), written for the time interval $\left[\nu_{k-1},\nu_{k}\right]$, result in the following system: (20)$$\begin{array}{cc} \frac{\gamma_{k}}{\beta_{k}}\ln\left(\sigma_{k-1}\right)+C_{k} & =\sigma_{k-1}+\lambda_{k-1}, \end{array}$$(21)$$\begin{array}{cc} \frac{\gamma_{k}}{\beta_{k}}\ln\left(\sigma_{k}\right)+C_{k} & =\sigma_{k}+\lambda_{k}, \end{array}$$
for k=2,3,…,m.

The system of Equations (Equation 20) and (Equation 21), after excluding $C_{k}$, gives a formula for the ratio $\frac{\gamma_{k}}{\beta_{k}}$:(22)$$\frac{\gamma_{k}}{\beta_{k}}=\frac{\sigma_{k}-\sigma_{k-1}+\lambda_{k}-\lambda_{k-1}}{\ln\left(\frac{\sigma_{k}}{\sigma_{k-1}}\right)}\;\cdot$$

Substituting the ratio (Equation 22) into Equations (Equation 11) and (Equation 12) gives the following expressions for the basic and effective reproduction numbers $R_{0,k}$ and $R_{e,k}$:(23)$$R_{0,k}=\frac{\beta_{k}}{\gamma_{k}}\;N=\frac{\ln\left(\frac{\sigma_{k}}{\sigma_{k-1}}\right)}{\sigma_{k}-\sigma_{k-1}+\lambda_{k}-\lambda_{k-1}}\;N,\;\;R_{e,k}=\frac{\beta_{k}}{\gamma_{k}}\;\sigma_{k}=\frac{\ln\left(\frac{\sigma_{k}}{\sigma_{k-1}}\right)}{\sigma_{k}-\sigma_{k-1}+\lambda_{k}-\lambda_{k-1}}\;\sigma_{k},$$
respectively, on the time interval $\left[\nu_{k-1},\nu_{k}\right]$ for k=2,3,…,m.

If the number of susceptible individuals at two consecutive time moments $\nu_{k-1}$ and $\nu_{k}$ is the same, i.e., $\sigma_{k-1}=\sigma_{k}$, then the number of newly infected individuals for this period is zero, and the reproduction number is zero. In other words, the infection is starting to decline.

### 3.2. Estimating β/γ as a Solution of BVP for Equation (Equation 7)

Equation (Equation 7) and the conditions (Equation 17) and (Equation 18), written for the time interval $\left[\nu_{k-1},\nu_{k}\right]$, result in the following system for k=2,3,…,m: (24)$$\begin{array}{cc} \frac{\beta_{k}}{\gamma_{k}}\rho_{k-1}+C_{k} & =-\ln\sigma_{k-1}, \end{array}$$(25)$$\begin{array}{cc} \frac{\beta_{k}}{\gamma_{k}}\rho_{k}+C_{k} & =-\ln\sigma_{k}. \end{array}$$

The system of Equations (Equation 24) and (Equation 25) gives a formula for the ratio $\frac{\beta_{k}}{\gamma_{k}}$:(26)$$\frac{\beta_{k}}{\gamma_{k}}=\frac{\ln\left(\frac{\sigma_{k-1}}{\sigma_{k}}\right)}{\rho_{k}-\rho_{k-1}}\;\cdot$$

Substituting the ratio (Equation 26) into Equations (Equation 11) and (Equation 12) gives the expressions for the basic and effective reproduction numbers $R_{0,k}$ and $R_{e,k}$ on the time interval $\left[\nu_{k-1},\nu_{k}\right]$ for k=2,3,…,m:(27)$$R_{0,k}=\frac{\beta_{k}}{\gamma_{k}}N=\frac{\ln\left(\frac{\sigma_{k-1}}{\sigma_{k}}\right)}{\rho_{k}-\rho_{k-1}}N,\;R_{e,k}=\frac{\beta_{k}}{\gamma_{k}}\sigma_{k}=\frac{\ln\left(\frac{\sigma_{k-1}}{\sigma_{k}}\right)}{\rho_{k}-\rho_{k-1}}\sigma_{k}.$$

If the number of removed individuals, i.e., $\rho_{k-1}$ and $\rho_{k}$, for two consecutive time moments $\nu_{k-1}$ and $\nu_{k}$ is the same ($\rho_{k-1}=\rho_{k}$), then the number of newly infected individuals for this period is zero, and the reproduction ratio is also zero. In other words, the infection is diminishing.

### 3.3. Identifying $R_{0}\left(t\right)$ and $R_{e}\left(t\right)$ from Equation (Equation 5) Using LSM

Let the dataset for the values of *S* and *I* ($\sigma_{k}$ and $\lambda_{k}$, respectively) be divided into subsets of fixed length of *P* days, as shown in Figure 2.

Following the LSM, we construct the following functions:(28)$$\Psi \left(\frac{\gamma_{k}}{\beta_{k}},C_{k}\right)=\sum_{l=k-P}^{k}\mu_{l}{\left(\frac{\gamma_{k}}{\beta_{k}}\ln\left(\sigma_{l}\right)+C_{k}-\sigma_{l}-\lambda_{l}\right)}^{2}$$
for k=P+1,P+2,…,m. Here, $\mu_{l}$ is the weight coefficient of the corresponding nodes. The necessary conditions for the minimization of the function Ψ with respect to $\frac{\gamma_{k}}{\beta_{k}}$ and $C_{k}$ are as shown:(29)$$\frac{\partial \Psi }{\partial \frac{\gamma_{k}}{\beta_{k}}}=0,\;\frac{\partial \Psi }{\partial C_{k}}=0.$$

Let us introduce the following notations:(30)$$\begin{array}{c} A_{11}=\sum_{l=k-P}^{k}\mu_{l}{\left(\ln(\sigma_{l})\right)}^{2},\;A_{12}=\sum_{l=k-P}^{k}\mu_{l}\ln\left(\sigma_{l}\right),\;A_{22}=\sum_{l=k-P}^{k}\mu_{l}, \end{array}$$(31)$$\begin{array}{c} B_{1}=\sum_{l=k-P}^{k}\mu_{l}\left(\sigma_{l}+\lambda_{l}\right)\ln\left(\sigma_{l}\right),\;B_{2}=\sum_{l=k-P}^{k}\mu_{l}\left(\sigma_{l}+\lambda_{l}\right). \end{array}$$

The necessary conditions (Equation 29) become a system of two linear equations for the unknowns $\frac{\gamma_{k}}{\beta_{k}}$ and $C_{k}$:(32)$$A_{11}\;\frac{\gamma_{k}}{\beta_{k}}+A_{12}\;C_{k}=B_{1},\;A_{12}\;\frac{\gamma_{k}}{\beta_{k}}+A_{22}\;C_{k}=B_{2}.$$

The system (Equation 32) gives the values of the ratio $\frac{\gamma_{k}}{\beta_{k}}$ and $C_{k}$:(33)$$\frac{\gamma_{k}}{\beta_{k}}=\frac{A_{22}B_{1}-A_{12}B_{2}}{A_{11}A_{22}-A_{12}^{2}}\cdot$$

Then, the reproduction numbers become the following:(34)$$R_{0,k}=\frac{\beta_{k}}{\gamma_{k}}\;N=\frac{A_{11}A_{22}-A_{12}^{2}}{A_{22}B_{1}-A_{12}B_{2}}\;N,\;R_{e,k}=\frac{\beta_{k}}{\gamma_{k}}\;\sigma_{k}=\frac{A_{11}A_{22}-A_{12}^{2}}{A_{22}B_{1}-A_{12}B_{2}}\;\sigma_{k}.$$

Note that, in the case when P=1, the formulas in (Equation 34) are equivalent to the corresponding formulas in (Equation 23).

### 3.4. Identifying $R_{0}\left(t\right)$ and $R_{e}\left(t\right)$ from Equation (Equation 7) Using LSM

Similarly, following the LSM, we construct the following functions:(35)$$\Phi \left(\frac{\beta_{k}}{\gamma_{k}},C_{k}\right)=\sum_{l=k-P}^{k}\mu_{l}{\left(\frac{\beta_{k}}{\gamma_{k}}\rho_{l}+\ln\sigma_{l}+C_{k}\right)}^{2}\;,$$
for k=P+1,P+2,…,m. Here, $\mu_{l}$ is the weight coefficient of the corresponding nodes. The necessary conditions for the minimization of the function Φ with respect to $\frac{\beta_{k}}{\gamma_{k}}$ and $C_{k}$ are below:(36)$$\frac{\partial \Phi }{\partial \frac{\beta_{k}}{\gamma_{k}}}=0,\;\frac{\partial \Phi }{\partial C_{k}}=0.$$

After introducing the following notations:(37)$$\begin{array}{c} D_{11}=\sum_{l=k-P}^{k}\mu_{l}\rho_{l}^{2},\;D_{12}=\sum_{l=k-P}^{k}\mu_{l}\rho_{l},\;D_{22}=\sum_{l=k-P}^{k}\mu_{l}, \end{array}$$(38)$$\begin{array}{c} E_{1}=\sum_{l=k-P}^{k}\mu_{l}\rho_{l}\ln\left(\sigma_{l}\right),\;E_{2}=\sum_{l=k-P}^{k}\mu_{l}\ln\left(\sigma_{l}\right), \end{array}$$
the necessary conditions (Equation 36) become a system of two linear equations for the unknowns $\frac{\beta_{k}}{\gamma_{k}}$ and $C_{k}$:(39)$$D_{11}\;\frac{\beta_{k}}{\gamma_{k}}+D_{12}\;C_{k}=-E_{1},\;D_{12}\;\frac{\beta_{k}}{\gamma_{k}}+D_{22}\;C_{k}=-E_{2}.$$

The system (Equation 39) yields the values of the ratio $\frac{\beta_{k}}{\gamma_{k}}$
(40)$$\frac{\beta_{k}}{\gamma_{k}}=\frac{D_{12}E_{2}-D_{22}E_{2}}{D_{12}^{2}-D_{11}D_{22}}\cdot$$

Then, the reproduction numbers become the following:(41)$$R_{0,k}=\frac{\beta_{k}}{\gamma_{k}}\;N=\frac{D_{12}E_{2}-D_{22}E_{2}}{D_{12}^{2}-D_{11}D_{22}}\;N,\;R_{e,k}=\frac{\beta_{k}}{\gamma_{k}}\;\sigma_{k}=\frac{D_{12}E_{2}-D_{22}E_{2}}{D_{12}^{2}-D_{11}D_{22}}\;\sigma_{k}.$$

Note that the LSM solution is equivalent to the BVP solution if P=1.

## 4. Results

We applied the method for reproduction number identification to real COVID-19 data, available in [44], for the countries of the United States and Canada, and the states of Georgia, Texas, and Louisiana, for a period ending on 4 January 2023.

We conducted simulations using the approaches described in Section 3. The solutions to the BVP are very sensitive to random perturbations in data values, as seen in Figure 3, while Figure 4 presents the identified values using the LSM method.

The numerically identified reproduction numbers using BVP and LSM follow the same trend, although the LSM method over 7 or more days helps reduce the effect of reported data containing errors, e.g., due to not being reported daily. This is why the basic and effective reproduction numbers are identified over 3 different time periods: 7 days; 14 days; and 21 days. The results using LSM for the solutions of (Equation 6) and (Equation 8) are identical. For this reason, herein we present the results obtained using Equation (Equation 6).

The results obtained from this work show an interesting relationship between the number of currently infectious individuals (active COVID-19 cases reported in [44]) and the effective reproduction number, namely:


*The local maximum/minimum values of the effective reproduction number $R_{e}\left(t\right)$ occur approximately three weeks before the local maximum/minimum values of the number of currently infectious individuals I(t).*


This observed behavior is due to the fact that the reproduction number influences the rate of spread of an epidemic. This means that, if the reproduction number starts to increase/decrease, then this causes the infectious disease spread to rise/decline. The obtained numerical results are in agreement with this behavior. Finding mathematical proof of this statement is worth pursuing.

We illustrate the observations for the above relation by comparing the COVID-19 data for the daily infectious individuals I(t) and the computed effective reproduction number $R_{e}\left(t\right)$, based on a 21-day period, for the selected countries and states.

### 4.1. United States

The obtained results for the basic reproduction number $R_{0}\left(t\right)$ and the effective reproduction number $R_{e}\left(t\right)$ are given in Figure 4. The values for the last few months are lower compared to those prior to February 2022.

Figure 5 compares the local maximum values of the number of currently infectious individuals and the corresponding local maxima of the identified effective reproduction number $R_{e}$. This observation provides us with an instrument for predicting the time of the processes of decline in an epidemic wave.

The shapes of the graphs for the currently infectious people and the effective reproduction number $R_{e}$ are similar. To demonstrate the difference, we present a graph with two curves, one for the number of infectious individuals, and another for the effective reproduction number $R_{e}$, scaled by 1,500,000.

### 4.2. Canada

Figure 6 presents the obtained results for $R_{0}\left(t\right)$ and $R_{e}\left(t\right)$. As expected, the shapes of the graphs for the currently infectious population and the effective reproduction number $R_{e}\left(t\right)$ are similar. Figure 7 compares the number of currently infectious individuals I(t) and the effective reproduction number $R_{e}\left(t\right)$, scaled by 50,000. Figure 7 confirms the connection between the local maximum values of the number of currently infectious people and the corresponding local maxima of the effective reproduction number.

### 4.3. Texas

The results for Texas, shown in Figure 8 and Figure 9, show a similar tendency. Figure 9 clearly illustrates the connection between local maximum values of the number of currently infectious individuals and the corresponding local maxima of the effective reproduction number $R_{e}\left(t\right)$, multiplied by 150,000.

### 4.4. Georgia and Louisiana

Georgia and Louisiana have relatively small populations compared to Texas, Canada, and the United States. Figure 10 presents the results for $R_{e}\left(t\right)$, while Figure 11 displays I(t) and 15,000 $R_{e}\left(t\right)$ for the states of Georgia and Louisiana. These results clearly confirm the claim that the local maximum values of the effective reproduction number $R_{e}\left(t\right)$ occur a few weeks before the local maximum values of the number of currently infectious individuals I(t).

## 5. Discussion

This work presents an instrument for the study and analysis of the spread of infectious diseases within the scope of compartmental SIR-type models. We have developed efficient methods for determining the time-dependent basic and effective reproduction numbers, $R_{0}\left(t\right)$ and $R_{e}\left(t\right)$, through an inverse problem approach. We have applied the developed method to publicly reported COVID-19 data in [44] for Canada and the United States, as well as for the states of Texas, Georgia, and Louisiana. The method uses direct integration of equations resulting from the SIR system and allows for time-dependent infection and recovery rates, thus enabling the results to capture the changes in the reproduction numbers.

We applied the proposed methods to obtain explicit expressions for the time-dependent basic and effective reproduction numbers, $R_{0}\left(t\right)$ and $R_{e}\left(t\right)$, respectively. The inverse problem for the time-dependent reproduction numbers uses available data for the number of susceptible and infectious individuals. We developed two approaches for the identification of the reproduction numbers, namely, the Boundary Value Problem (BVP) and Least Squares Method (LSM), for the derived equations.

Next, we presented and analyzed the numerical results for the selected counties and states. The obtained results demonstrate a relationship between the values of the number of infected individuals I(t) and the scaled effective reproduction number $R_{e}\left(t\right)$. The curves follow a distinct pattern, where the local maxima and minima differ by a few weeks, with the effective reproduction number extrema preceding those of the reported data for I(t). In the future, we aim to provide a mathematical proof of the observed correlation between the maximum (and minimum) values of the number of currently infectious individuals and those of the effective reproduction number.

Mathematical modeling of infectious diseases has many known limitations. This study involves assumptions, such as that the reported data include all the cases; that the population under study is well mixed; and that infectious individuals leave the I(t) class and move directly into the R(t) class, although it is now known that some who recovered from COVID-19 individuals can be infected again. The proposed model does not consider asymptomatic and presymptomatic transmissions, which play an important role in spreading an epidemic [45]. Furthermore, numerous symptomatic cases of COVID-19 are unreported. Neglecting to account for these cases leads to artificially inflated transmission rates and, consequently, higher reproduction numbers.

We tested the developed algorithms against reported data for the active cases from [44]. However, we acknowledge that the assumptions for the reported data pose a limitation, since they are not exactly true for COVID-19. Negative tests do not guarantee that individuals are not currently infectious, and there are diagnostic delays (especially with the omicron variant) which may not be detected on the widely used lateral flow test kits until five or six days later, according to [46]. The SIR-type models assume mixing of the population, lack of reinfection, a constant population, and accuracy and completeness of the reported data. Therefore, the models may not capture all the complexities of the pandemic, and the results should be interpreted with caution.

## 6. Conclusions

This work presents a novel method for modeling epidemics by using an inverse problem approach to efficiently estimate time-dependent reproduction numbers. We solve the inverse problem by defining the boundary value problem (BVP) and the least squares method (LSM) problems, using datasets of the examined population obtained from publicly available COVID-19 data. The estimated values of the reproduction numbers provide a tool for predicting the spread of infectious diseases. Despite the limitations of the model and the data quality, this study presents a framework for analyzing the spread of infectious diseases. In particular, the observed relationship between the effective reproduction number and the infection data can provide insight into the behavior of the infection curve.

## Figures and Tables

**Figure 1 viruses-15-01352-f001:**
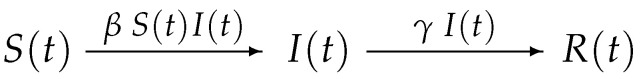
The SIR epidemic model.

**Figure 2 viruses-15-01352-f002:**
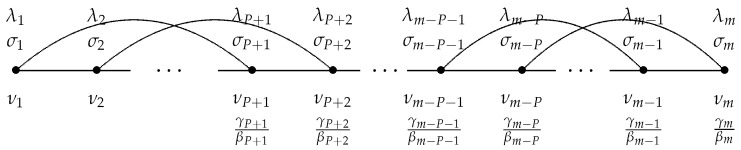
The time nodes and the subsets of fixed length of *P* days for identifying γ(t)/β(t).

**Figure 3 viruses-15-01352-f003:**
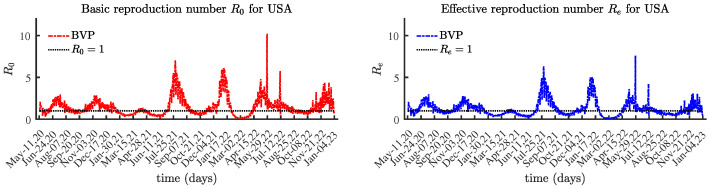
The basic and effective reproduction numbers for United States with BVP.

**Figure 4 viruses-15-01352-f004:**
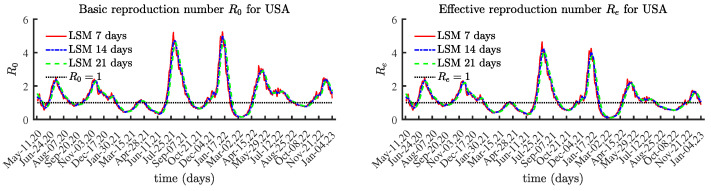
Basic and effective reproduction numbers for United States with LSM.

**Figure 5 viruses-15-01352-f005:**
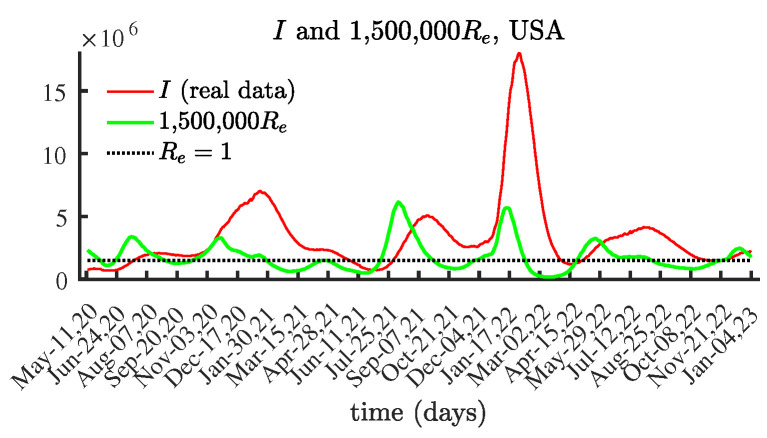
The number of currently infectious people I(t) and the effective reproduction number $R_{e}\left(t\right)$ scaled by 1,500,000, United States.

**Figure 6 viruses-15-01352-f006:**
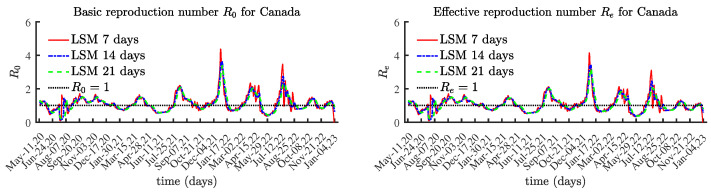
The basic and effective reproduction numbers for Canada.

**Figure 7 viruses-15-01352-f007:**
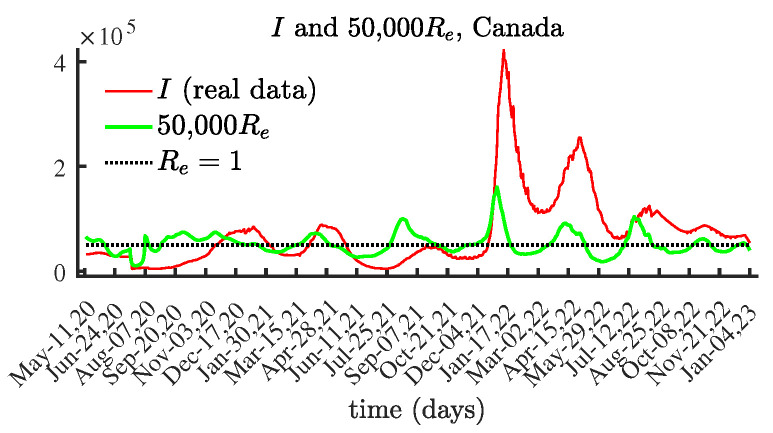
The number of currently infectious people I(t) and the effective reproduction number $R_{e}\left(t\right)$ scaled by 50,000, Canada.

**Figure 8 viruses-15-01352-f008:**
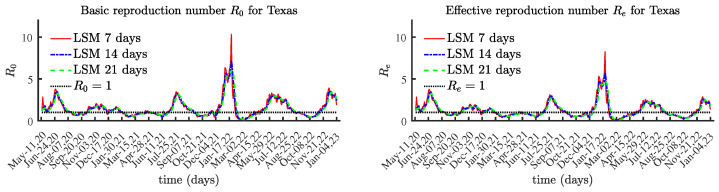
The basic and effective reproduction numbers for Texas.

**Figure 9 viruses-15-01352-f009:**
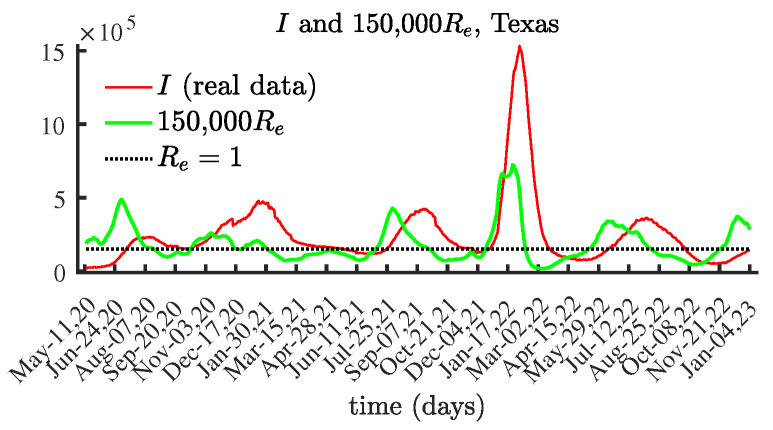
The number of currently infectious people I(t) and the effective reproduction number $R_{e}\left(t\right)$ scaled by 150,000, Texas.

**Figure 10 viruses-15-01352-f010:**
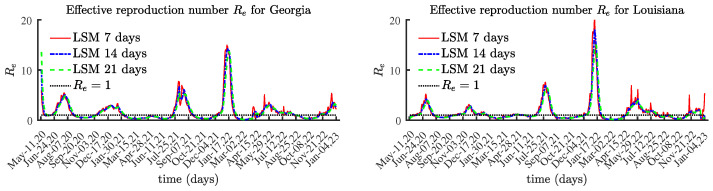
The effective reproduction number for Georgia and Louisiana.

**Figure 11 viruses-15-01352-f011:**
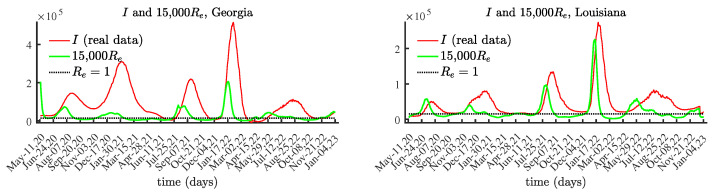
The number of currently infectious people I(t) and the effective reproduction number $R_{e}\left(t\right)$ scaled by 15,000, Georgia and Louisiana.

## Data Availability

All the data used in this work are available and published. The COVID-19 data used in this research are available at https://www.worldometers.info/coronavirus/.

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
