# Peer review of "Novel Approach for Identification of Basic and Effective Reproduction Numbers Illustrated with COVID-19"

_viruses, 2023, doi:10.3390/v15061352_

Round 1

Reviewer 1 Report

There are several mathematical models used to estimate the basic reproduction number (R0) for COVID-19. The specific focus on the SIR method is unclear, and as the understanding of COVID-19 has evolved, the model and approach to estimate R0 should evolve as well, i.e. to estimate it more accurately.

Specific comments:

1. The study title is vague and can be improved. Is the study meant to be a proposal for a new mathematical model to calculate R0? What type of study is this?

2. The style for intext citation is incomplete, e.g. "is defined in epidemiology, according to [2]". Please correct this.

3. Much of the methods contain unnecessary information, e.g. "An epidemic occurs if an infective individual introduced into a population of S(t) 75 susceptible individuals at the time moment t infects on average more than one other person. When the fraction of the population that is immune increases (because of vaccination or because of recovering from the disease) so much that Re(t) < 1, "herd immunity" has been achieved and the number of new cases occurring in the population will decrease to zero", that is likely already well known to scientists and readers of the journal. Please rewrite this section in a more succinct manner.

4. How exactly do you determine the "currently infective individuals"?

5. The assumptions and limitations of the model should be clearly stated. Models often assume homogeneity in the population, assuming that individuals have the same probability of contact, which is hardly the case in reality.

6. A single negative test is not a guarantee that someone is not currently infective, and there are diagnostic delays especially with the omicron variant, which may not be detected on the widely used lateral flow test kits until five or six days later (citation: pubmed.ncbi.nlm.nih.gov/36431077). These are important considerations that should be mentioned and discussed.

7. How does the proposed model account for asymptomatic and presymptomatic transmission? How can we estimate the probability that a COVID-19 infection is symptomatic?

8. Is there a particular reason why the model was applied to very recent data only? I would suggest applying the model to data from earlier in the pandemic, for example, during the alpha vs delta vs omicron variant era. The current estimates are likely underestimated given that seriously reduced testing capacity and reporting rate for COVID-19 infections. Most countries also only rely on the antigen test for diagnosis nowadays, which is much less accurate than the PCR test.

9. What is the strength of the proposed approach compared to the other proposed mathematical models? This seems like a purely illustrative paper of the SIR method with no novelty. There are many different models in literature that consider incorporate different features of the epidemic such as asymptomatic and presymptomatic transmission, superspreading, or heterogeneity in susceptibility/immunity. Bayesian methods in particular use statistical techniques to estimate R0 based on observed data and prior knowledge. These methods incorporate uncertainty common in the early stages of the pandemic and can update estimates as new data becomes available. Bayesian models can be adapted to various epidemiological models, such as SIR or SEIR, and allow for the incorporation of more complex factors.

10. "The developed method can be extended to include other factors, which play important roles in disease propagation" - this should be demonstrated in your results!

11. Suggest having a separate conclusions section to reiterate the key findings and areas for future work.

There are some grammatical and language errors that require correction.

Author Response

Comments and Suggestions for Authors

There are several mathematical models used to estimate the basic reproduction number (R0) for COVID-19. The specific focus on the SIR method is unclear, and as the understanding of COVID-19 has evolved, the model and approach to estimate R0 should evolve as well, i.e. to estimate it more accurately.

We agree with the comment. We emphasized throughout the text that the basic reproduction number R0 in the classical SIR model is a constant; on the other hand, the effective reproduction number is a function of time due to S being time-dependent. Data shows that the transmission and removal rates, and R0 are not constants in the case of COVID-19 – they are time-dependent too.

Therefore, our approach considers the ratio beta/gamma as time-dependent and this is a novel approach, which extends upon SIR.  We aim to find beta/gamma as a function of time t. 

Specific comments:

  1. The study title is vague and can be improved. Is the study meant to be a proposal for a new mathematical model to calculate R0? What type of study is this?

Yes, the title could improve. We changed it to: “Novel Approach for Identification of Basic and Effective Reproduction Numbers Illustrated with COVID-19”

  1. The style for intext citation is incomplete, e.g. "is defined in epidemiology, according to [2]". Please correct this.

Corrected.

  1. Much of the methods contain unnecessary information, e.g. "An epidemic occurs if an infective individual introduced into a population of S(t) 75 susceptible individuals at the time moment t infects on average more than one other person. When the fraction of the population that is immune increases (because of vaccination or because of recovering from the disease) so much that Re(t) < 1, "herd immunity" has been achieved and the number of new cases occurring in the population will decrease to zero", that is likely already well known to scientists and readers of the journal. Please rewrite this section in a more succinct manner.

We removed this paragraph.

  1. How exactly do you determine the "currently infective individuals"?

We use reported data for the active cases from: https://www.worldometers.info/coronavirus/.   We clarified this in the Results section “currently infective individuals (active COVID-19 cases reported in \cite{worldometer})”

  1. The assumptions and limitations of the model should be clearly stated. Models often assume homogeneity in the population, assuming that individuals have the same probability of contact, which is hardly the case in reality.

We included statements regarding the assumptions and limitations in the Discussion section. 

  1. A single negative test is not a guarantee that someone is not currently infective, and there are diagnostic delays especially with the omicron variant, which may not be detected on the widely used lateral flow test kits until five or six days later citation: pubmed.ncbi.nlm.nih.gov/36431077 ). These are important considerations that should be mentioned and discussed.

This consideration is true. We mentioned and discussed it in the manuscript.

  1. How does the proposed model account for asymptomatic and presymptomatic transmission? How can we estimate the probability that a COVID-19 infection is symptomatic?

Our model does not consider asymptomatic and presymptomatic transmission. We tested the developed algorithms reported data for the active cases from: https://www.worldometers.info/coronavirus/.

We stated this in the limitations of the study.

  1. Is there a particular reason why the model was applied to very recent data only? I would suggest applying the model to data from earlier in the pandemic, for example, during the alpha vs delta vs omicron variant era. The current estimates are likely underestimated given that seriously reduced testing capacity and reporting rate for COVID-19 infections. Most countries also only rely on the antigen test for diagnosis nowadays, which is much less accurate than the PCR test.

Good comment / suggestion. There is no particular reason for not applying the model to data from earlier in the pandemic. We extended the period to start sometime May 2020, ending is still January 4.

  1. What is the strength of the proposed approach compared to the other proposed mathematical models? This seems like a purely illustrative paper of the SIR method with no novelty. There are many different models in literature that consider incorporate different features of the epidemic such as asymptomatic and presymptomatic transmission, superspreading, or heterogeneity in susceptibility/immunity. Bayesian methods in particular use statistical techniques to estimate R0 based on observed data and prior knowledge. These methods incorporate uncertainty common in the early stages of the pandemic and can update estimates as new data becomes available. Bayesian models can be adapted to various epidemiological models, such as SIR or SEIR, and allow for the incorporation of more complex factors.

We clarified (in several places such as the abstract, introduction, discussion, and conclusion) that the proposed approach assumes time-dependent basic reproduction number and develops an efficient method for identifying the time-dependent basic and effective reproduction numbers. The method is not statistical, it uses inverse problem for solving differential equations using additional data for the number of reported active cases. 

  1. "The developed method can be extended to include other factors, which play important roles in disease propagation" - this should be demonstrated in your results!

We removed this sentence.

  1. Suggest having a separate conclusions section to reiterate the key findings and areas for future work.

We added a separate conclusion section as suggested.

Comments on the Quality of English Language

There are some grammatical and language errors that require correction.

We have read the text once again and performed minor edits of the English language.

Reviewer 2 Report

In this study the Authors present the application of one of the compartmental models, namely SIR, applied for the COVID-19 pandemics. While the study is concise, it also requires some revisions listed below. Also, due to the quite limited amount of new information I would rather suggest publication in MDPI’s COVID than in Viruses.

Lines 3 and 30 – „SIR” abbreviations must be explained. Although it is well known to those using compartmental models in epidemiology, it should be noted that some of the Readers of Viruses are not experts in this particular field.

In the introduction the variations of the basic SIR model must be explained, such as i.e. SIS, SIRD, SIRV and MSEIR.

Line 42, β is also lower than 1. The same applies to γ in line 45.

The aim of the study is not stated clearly.

Lines 186-188, this observation should be explained and discussed.

Figure 7, left panel, the results exceed the scale.

The SIR model, used in this study, is the most basic one. Why the Authors didn’t try to compare this model to the other, more advanced ones?

Author Response

Comments and Suggestions for Authors

In this study the Authors present the application of one of the compartmental models, namely SIR, applied for the COVID-19 pandemics. While the study is concise, it also requires some revisions listed below. Also, due to the quite limited amount of new information I would rather suggest publication in MDPI’s COVID than in Viruses. The aim of the study is not stated clearly.

We revised the aim of the study paragraph to state it clearly: “The present work proposes a novel approach for the identification of the time-dependent reproduction numbers. It extends the SIR-type model by assuming time-dependent infection and recovery rates [40,41], required for the estimation of the reproduction numbers. The method is based on direct integration of the SIR system and an inverse problem approach for solving the resulting equation for the ratio of the infection and recovery rates. The evaluation of the time-dependent reproduction numbers over long time periods gives a tool for investigating the wave formation of the infection curve for the COVID-19 epidemic. This study examines and shows how the time-dependent effective reproduction number compares to the number of infective individuals in time and gives a novel approach of predicting epidemic waves.”

Lines 186-188, this observation should be explained and discussed.

We added the following explanation and discussion:

“This observed behavior is due to the fact that the reproduction number influences the rate of spread of an epidemic. This means that if the reproduction number starts to increase / decrease, then this causes the infectious disease spread to rise / decline. The obtained numerical results are in agreement with this behavior. Finding a mathematical proof of this statement is worth doing.”

Lines 3 and 30 – „SIR” abbreviations must be explained. Although it is well known to those using compartmental models in epidemiology, it should be noted that some of the Readers of Viruses are not experts in this particular field.

We included the definition of the acronym SIR in the abstract and the first time using it.

In the introduction the variations of the basic SIR model must be explained, such as i.e. SIS, SIRD, SIRV and MSEIR.

We added a paragraph explaining SIS, SIRD, SIRV and MSEIR in the introduction.

Line 42, β is also lower than 1. The same applies to γ in line 45.

We added this.

Figure 7, left panel, the results exceed the scale.

Fixed it.

The SIR model, used in this study, is the most basic one. Why the Authors didn’t try to compare this model to the other, more advanced ones?

The time-dependent SIR model we use in this study is compared to other models. We also clarified the difference between the most basic SIR model and the model we considered (which considers time-dependent transmission and recovery rates).    

Reviewer 3 Report

1) Abstract: This research is focused on a special numerical technique for identifying the effective and 1basic reproduction numbers Re and R0 for long term epidemics. The method is based on a direct  integration of the SIR system of ordinary differential equations and the least squares method. The  simulations are based on official COVID-19 data for the United States and Canada, and the states of Georgia, Texas, and Louisiana, for a period of two years. The results demonstrate how well the  method simulates the dynamics. The obtained results from this research show an interesting relation  between the number of currently infective individuals and the effective reproduction number, which is a useful tool for predicting the epidemic dynamics. Please improve the abstract regarding the results and the conclusions.

2) 1. Introduction L11-14. Infectious diseases have became a particularly important subject of study during  the last several years due to the COVID-19 epidemic. Worldwide daily reports provided  scientists with data utilized to develop new approaches for studying infectious diseases. Identifying outbreaks at early stages is of great concern to public health authorities [1]. The Authors should cite some more of the important studies in the area on this topic, and discuss in the Discussion Section how their study adds to the current data. I suggest:

A- Different Methods to Improve the Monitoring of Noninvasive Respiratory Support of Patients with Severe Pneumonia/ARDS Due to COVID-19: An Update. J Clin Med. 2022 Mar 19;11(6):1704. doi: 10.3390/jcm11061704. 

B-  Cardiovascular and Neurological Complications of COVID-19: A Narrative Review. J Clin Med. 2023 Apr 12;12(8):2819. doi: 10.3390/jcm12082819.

C- Hyperinflammatory Response in COVID-19: A Systematic Review. Viruses 202315, 553. https://doi.org/10.3390/v15020553.

3) 1. Introduction. L30-34. The present work uses a SIR-type model focusing on novel approaches for the iden- tification of the time-dependent effective reproduction number and the prediction of the wave formation of the infection curve for the COVID-19 epidemic. SIR models displayed compelling results especially during the early period of the pandemic [14–27]. Recent works implement various SIR-type models to investigate the COVID-19 infectious disease dynamics [7,28–30].  

Please, improve the description of study aim and underline the novelty of this observation.

4) 2. The SIR model for the spread of an infectious disease. L 38. It categorizes individuals as Susceptible, Infectious and  Recovered (SIR). Please explain all the acronyms the first time they are mentioned.

5) 4. Results L175-179. We applied the method for reproduction number identification to real COVID-19 data, available in [36], for the countries United States and Canada, and the states of Georgia, Texas, and Louisiana for a period of approximately two years ending on January 4, 2023. The basic and effective reproduction numbers are identified over three different time  periods: 7 days; 14 days; and 21 days. Please underline in the text the most important statisticaly signicant results to support the observations.

6) L 243-248. The developed method can be extended to include other factors, which play important  roles in disease propagation. We also acknowledge that the assumption for the reported data including all the cases is a limitation, because it is not exactly true for COVID-19. Regardless of the limitations of the model, the data quality and the discussed results, our work provides a framework for analysis of the infectious disease spread. In particular, the  observed relation between the effective reproduction number and the infectives data for  the selected regions gives insights into the behavior of the infection curve. Please, clarify the limitations of the study.

7) Please, insert a brief paragraph regarding the conclusions.

The manuscript is quite well written, I suggest minor edits of the English language

Author Response

Comments and Suggestions for Authors

1) Abstract: This research is focused on a special numerical technique for identifying the effective and basic reproduction numbers Re and R0 for long term epidemics. The method is based on a direct integration of the SIR system of ordinary differential equations and the least squares method. The simulations are based on official COVID-19 data for the United States and Canada, and the states of Georgia, Texas, and Louisiana, for a period of two years. The results demonstrate how well the method simulates the dynamics. The obtained results from this research show an interesting relation between the number of currently infective individuals and the effective reproduction number, which is a useful tool for predicting the epidemic dynamics.

Please improve the abstract regarding the results and the conclusions.

We have added the following sentence to the abstract:

“For all conducted experiments, the local maximum (and minimum) values of the time-dependent effective reproduction number occur approximately three weeks before the local maximum (and minimum) values of the number of currently infective individuals.”

2) 1. Introduction L11-14. Infectious diseases have became a particularly important subject of study during the last several years due to the COVID-19 epidemic. Worldwide daily reports provided scientists with data utilized to develop new approaches for studying infectious diseases. Identifying outbreaks at early stages is of great concern to public health authorities [1].

The Authors should cite some more of the important studies in the area on this topic, and discuss in the Discussion Section how their study adds to the current data. I suggest:

A- Different Methods to Improve the Monitoring of Noninvasive Respiratory Support of Patients with Severe Pneumonia/ARDS Due to COVID-19: An Update. J Clin Med. 2022 Mar 19;11(6):1704. doi: 10.3390/jcm11061704.

B-  Cardiovascular and Neurological Complications of COVID-19: A Narrative Review. J Clin Med. 2023 Apr 12;12(8):2819. doi: 10.3390/jcm12082819.

C- Hyperinflammatory Response in COVID-19: A Systematic Review. Viruses 2023, 15, 553. https://doi.org/10.3390/v15020553.

We examined the above publications on the topic and some other recent studies, and discussed them.

3) 1. Introduction. L30-34. The present work uses a SIR-type model focusing on novel approaches for the identification of the time-dependent effective reproduction number and the prediction of the wave formation of the infection curve for the COVID-19 epidemic. SIR models displayed compelling results especially during the early period of the pandemic [14–27]. Recent works implement various SIR-type models to investigate the COVID-19 infectious disease dynamics [7,28–30]. 

Please, improve the description of study aim and underline the novelty of this observation.

We revised the description of study aim to improve it and underline the novelty of this observation:

“The present work proposes a novel approach for the identification of the time-dependent reproduction numbers. It extends the SIR-type model by assuming time-dependent infection and recovery rates [40,41], required for the estimation of the reproduction numbers. The method is based on direct integration of the SIR system and an inverse problem approach for solving the resulting equation for the ratio of the infection and recovery rates. The evaluation of the time-dependent reproduction numbers over long time periods gives a tool for investigating the wave formation of the infection curve for the COVID-19 epidemic. This study examines and shows how the time-dependent effective reproduction number compares to the number of infective individuals in time and gives a novel approach of predicting epidemic waves.”

4) 2. The SIR model for the spread of an infectious disease. L 38. It categorizes individuals as Susceptible, Infectious and Recovered (SIR). Please explain all the acronyms the first time they are mentioned.

We included the definition of the acronym SIR in the abstract and the first time using it.

5) 4. Results L175-179. We applied the method for reproduction number identification to real COVID-19 data, available in [36], for the countries United States and Canada, and the states of Georgia, Texas, and Louisiana for a period of approximately two years ending on January 4, 2023. The basic and effective reproduction numbers are identified over three different time periods: 7 days; 14 days; and 21 days.

Please underline in the text the most important statistically significant results to support the observations.

We explained why LSM with at least 7 days was a necessary choice. We included a figure showing the results with the BVP for United States:

“We conducted simulations using the approaches described in Section 3. The solutions of the BVP are very sensitive to random perturbations in data values, as seen in Figure 3, while Figure 4 presents the identified values with the LSM method.

The numerically identified reproduction numbers using BVP and LSM follow the same trend although the LSM method over 7 or more days helps reduce the effect of reported data containing errors, e.g., due to not being reported daily. This is why the basic and effective reproduction numbers are identified over three different time periods: 7 days; 14 days; and 21 days. The results using LSM for the solutions of (6) and (8) are identical. For this reason, here we present the results obtained with equation (6).”

6) L 243-248. The developed method can be extended to include other factors, which play important roles in disease propagation. We also acknowledge that the assumption for the reported data including all the cases is a limitation, because it is not exactly true for COVID-19. Regardless of the limitations of the model, the data quality and the discussed results, our work provides a framework for analysis of the infectious disease spread. In particular, the observed relation between the effective reproduction number and the infectives data for the selected regions gives insights into the behavior of the infection curve.

Please, clarify the limitations of the study.

We added text in the Discussion section to clarify the limitations:

“Mathematical modeling of infectious diseases has many known limitations. This study involves assumptions such as: the reported data include all the cases; the population under study is constant and well mixed; infectious individuals leave the $I(t)$ class and move directly into the $R(t)$ class, although it is now known that some recovered from COVID-19 individuals can be infected again. The proposed model does not consider asymptomatic and presymptomatic transmission [45]. We tested the developed algorithms against reported data for the active cases from [44]. We acknowledge that for all considered case-studies, the assumptions for the reported data impose a limitation, because the assumptions are not exactly true for COVID-19. According [46], negative tests do not guarantee that individuals are not currently infective, and that there are diagnostic delays (especially with the omicron variant), which may not be detected on the widely used lateral flow test kits until five or six days later.”

7) Please, insert a brief paragraph regarding the conclusions.

We added a conclusion.  

Comments on the Quality of English Language

The manuscript is quite well written, I suggest minor edits of the English language

We have read the text once again and performed minor edits of the English language.

Round 2

Reviewer 1 Report

1. The abstract should also include the key statistics and findings.

2. Several of the keywords chosen for the paper are too long and inappropriate, e.g. "infectious disease analysis and predictions". Please revise the keywords.

3. The style of intext citation is still incorrect at many points of the manuscript, e.g. "active COVID-19 cases reported in [44]". Please do a close edit of the manuscript.

4. Apart from merely stating that the proposed model does not consider asymptomatic and presymptomatic transmission, it is important to review the literature in this area, e.g. discuss the percentage of such cases and the resulting implications of omission.

5. For the Figures, it would be better to plot 21-day moving averages for the x axis to smoothen out the peaks and valleys.

Edits for style and language still necessary.

Reviewer 2 Report

The Authors have improved their manuscript and I find this version acceptable. However, for the future, please indicate the changes done in the manuscript suring revision by using different color of the font, i.e. red one.

Few typos and grammar mistakes are still present.

Round 3

Reviewer 1 Report

Thank you for the replies.

Ok.

Reviewer 2 Report

Current (revised) version can be accepted for publication.